# Impact of Rearing Duration on Nutritional Composition, Flavor Characteristics, and Physical Properties of Asian Swamp Eel (*Monopterus albus*)

**DOI:** 10.3390/foods14101685

**Published:** 2025-05-09

**Authors:** Yuning Zhang, Wentao Xu, Weiwei Lv, Quan Yuan, Hang Yang, Weiwei Huang, Wenzong Zhou

**Affiliations:** 1Eco-Environmental Protection Research Institute, Shanghai Academy of Agricultural Sciences, Shanghai 201403, China; ynzhang@saas.sh.cn (Y.Z.); wwlv1986@sina.com (W.L.); quanyuan2016@126.com (Q.Y.); yanghangqu@foxmail.com (H.Y.); 2Key Laboratory of Integrated Rice-Fish Farming, Ministry of Agriculture and Rural Affairs, Shanghai Academy of Agricultural Sciences, Shanghai 201403, China; 3Taizhou Institute of Agricultural Science, Jiangsu Academy of Agricultural Sciences, Taizhou 225300, China; xuwentao@jaas.ac.cn

**Keywords:** Asian eel, nutritional composition, texture analysis, flavor profiles

## Abstract

The Asian eel, a medicinal and edible species, lacks systematic research on age-related nutritional and flavor dynamics. To optimize breeding strategies and product differentiation, this study systematically investigated the nutritional composition, flavor profiles, and physical properties of Asian eel muscles across five distinct growth stages (1, 3, 7, 11, and 22 years). Results showed that unsaturated fatty acids increased with age, while ω-3/ω-6 ratios peaked in 1-year-old eels. The levels of hydrolyzed essential amino acids were higher in the 3–11-year-old groups, contrasting with higher free amino acids in 1- and 22-year-old eels. Texture declined in hardness/chewiness but improved in resilience with age, linked to muscle fiber density and diameter. One–three-year-old eels exhibited compact muscle fibers and superior texture, while 7–22-year groups demonstrated functional lipid profiles (high docosahexaenoic acid and γ-aminobutyric acid, low cholesterol). These findings highlight age-specific quality traits: 1–3-year-old eels are suitable for fresh consumption, 3–11-year groups offer bioactive benefits, and 22-year-old eels serve as premium functional ingredients. The study provides a scientific basis for targeted breeding and market segmentation to enhance the value of eel aquaculture.

## 1. Introduction

The Asian swamp eel (*Monopterus albus*) is an economical freshwater species. Due to its high protein content, low-fat composition, and richness in bioactive compounds, *M. albus* is valued in aquaculture and dietary therapy [1]. According to the China Fisheries Statistical Yearbook, the annual aquaculture production of eels has consistently exceeded 300,000 tons [2]. However, expectations for aquatic products have evolved significantly. Modern consumers increasingly prioritize not only guaranteed nutritional value but also enhanced flavor and taste profiles [1,3]. Consequently, the nutritional composition and sensory attributes of eel flesh have become critical factors in determining market competitiveness.

The nutritional and flavor characteristics of fish muscle are regulated by a combination of genetic [4], dietary [5], environmental [4], and ontogenetic factors [6,7]. Among these, age plays a pivotal role in modulating muscle quality by regulating lipid metabolism, protein dynamics, and muscle fiber architecture [8,9]. Studies indicate that age-dependent regulation is exemplified by shifts in fatty acid profiles across developmental stages [10]. In *Procambarus clarkii*, for instance, aging correlates with diminished fatty acid diversity and reduced total polyunsaturated fatty acids (PUFAs), while eicosapentaenoic acid (EPA) and arachidonic acid (ARA) levels exhibit marked elevation [11]. Parallel patterns emerge in *Culter alburnus*, where both fatty acid and amino acid profiles display age-specific variations [12]. Such metabolic divergence across growth phases drives significant age-related differences in amino acid composition and physiological demands [13]. In beluga (*Huso huso*), the total essential amino acids (ΣEAA) increase with age, a phenomenon likely linked to growth conditions such as diet composition and environmental factors [14]. Conversely, in American shad (*Alosa sapidissima*), nutrient allocation is prioritized to gonads during reproduction, leading to decreased amino acid reserves in muscle and liver [15]. These findings underscore the necessity of age-specific analyses to decode the biochemical determinants of fish meat quality.

Texture, a critical sensory attribute in aquatic products, is shaped by intrinsic factors (such as structural characteristics, chemical composition, color, fat content) and extrinsic variables (such as aquaculture practices, feed formulations, and pre- to post-slaughter processing) [16,17,18]. Numerous studies have revealed the impact of feeding methods on the texture of fish muscle. Wild specimens of most fish species—including sea bass (*Dicentrarchus labrax*) [16], Atlantic salmon (*Salmo salar*) [17], and blackspot seabream (*Pagellus bogaraveo*) [19]—exhibit superior hardness, crispness, and muscle quality compared to their farmed counterparts. In addition, dietary interventions further modulate texture profiles: supplementation with rapeseed oil and palm oil enhances muscle hardness, elasticity, and shear force, accompanied by histological evidence of enlarged and densely packed muscle fibers [20,21]. Mechanistically, dietary regimes influence muscle fiber diameter distribution and density [22], while molecular pathways such as FoxO1 signaling, AMPK-regulated glycogen/protein metabolism [23], and the gut–muscle axis [24] govern muscle mass development. Despite these advances, the age-dependent evolution of textural and nutritional traits in cultured fish remains underexplored.

As a benthic carnivorous fish with a unique life cycle (such as sex reversal), the Asian eel has been extensively studied in terms of breeding modes [25,26], nutritional components [27,28], and specific growth stages [25]. However, systematic investigations into the age-dependent dynamics of muscle nutritional and flavor characteristics remain limited, which affects breeding strategies and leads to low resource utilization efficiency. Therefore, this study systematically analyzes the dynamic changes in nutritional composition, flavor compounds, and textural properties of eel muscles across five developmental phases (1, 3, 7, 11, and 22 years). The findings are expected to provide actionable insights into precision breeding and market-driven product differentiation, thereby enhancing the sustainability and economic value of the eel aquaculture industry.

## 2. Materials and Methods

The Animal Ethics Committee of the Shanghai Academy of Agricultural Sciences approved all animal procedures under approval number SAASXM062438; the approval date was 7 August 2024.

### 2.1. Chemicals and Reagents

The 75% ethanol (chemical grade), 95% ethanol (analytical grade), 5% paraformaldehyde (chemical grade), hydrochloric acid (analytical grade), ammonia (analytical grade), pyrogallic acid (analytical grade), ether (analytical grade), sodium hydroxide (analytical grade), methanol (HPLC grade), acetonitrile (HPLC grade), heptane (HPLC grade), isooctane (HPLC grade), and sodium chloride (analytical grade) were purchased from Sangon Biotech (Shanghai, China).OPA-derived reagents and FMOC-derived reagents (analytical grade) were purchased from Sigma Corporation (Cream Ridge, NJ, USA). The standards of fatty acid methyl ester, amino acid, GABA, and cholesterol (Purity ≥ 99.9%) were purchased from the China National Institute for the Control of Pharmaceutical and Biological Products (Beijing, China). The triglyceride test kit was purchased from Nanjing Jiancheng Bioengineering Institute (Nanjing, China).

### 2.2. Experimental Management

The experimental site and fish were provided by the Zhuanghang Experimental Station of the Shanghai Academy of Agricultural Sciences. The aquaculture pond covers an area of 40 m^2^ and has a water depth of 0.6–0.8 m. Fish were adopted in net cages and fed a commercial diet, containing 43% crude protein and 7% crude lipids (Huisheng Biotechnology Co., Ltd., Wuhan, China) at 4:00 pm every day. Feeding protocols followed the standardized management model established in our laboratory’s prior studies to ensure uniform growth conditions [29]. During the experiment, the dissolved oxygen level exceeded 6 mg/L, the pH range was 7.5 to 8.0, the ammonia nitrogen level was below 0.2 mg/L, and the temperature range was 30 ± 2 °C.

### 2.3. Sample Collection

Randomly select 9 *M. albus* at the ages of 1, 3, 7, 11, and 22, and anesthetize them with MS-222 before sampling. Subsequently, the surface of *M. albus* was wiped with 75% ethanol to collect muscle samples, and samples were taken from both sides of the muscle using a disinfected surgical knife. Immediately determine the parameters of the back muscle texture. Muscle slices were fixed in 5% paraformaldehyde (PFA) for 24 h at 4 °C. The remaining samples were frozen using liquid nitrogen and then stored at −80 °C.

### 2.4. Experimental Detection

#### 2.4.1. Muscle Component Analysis

Muscle fatty acids were analyzed by a Shimadzu GC-2030AF gas chromatograph (Kyoto, Japan) according to the standard [30]. Muscle amino acid profiles (including hydrolyzed and free amino acids) were obtained using an Agilent 1260 HPLC system (Santa Clara, CA, USA), following the previous method [31]. This HPLC system is also used to detect the contents of GABA and cholesterol, according to the standards [32,33], respectively. Muscle triglyceride contents were detected using a Mindray UV visible spectrophotometer (Shenzhen, China), following the instructions of the Nanjing Jiancheng triglyceride kit (Nanjing, China).

#### 2.4.2. Muscle Texture, Slice Analysis

The muscle of *M. albus* was cut into sizes of 2 cm × 2 cm × 2 cm. Muscular textural parameters were tested by XTPlus texture analyzer (Stable Micro Systems, Croydon, UK) according to our previous research [26]. Texture indicators, such as muscle hardness, adhesion, cohesion, springiness, gumminess, chewiness, and Resilience are collected and normalized by software.

After fixation and dehydration, the muscle was embedded in paraffin and cut into 3–4 μm sections. Then, the slices were stained with hematoxylin–eosin (HE). We used a Nikon TS100 optical microscope with a magnification of 20× to measure the muscular histometry following the previous method [28]. Image Pro Plus 6.0 analysis software was used to check the number of muscle fibers and calculate the diameter of muscle fibers. Millimeters are used as the standard unit of measurement.

### 2.5. Data Analysis

After collecting the data, the experimental data were organized and statistically analyzed using IBM SPSS Statistics 22.0. All data are presented in the form of mean ± standard deviation (Mean ± SD). Apply one-way analysis of variance and conduct post hoc tests on the measured data using the LSD test and the Tamhane test.

## 3. Results

### 3.1. Content of Fatty Acid in Muscle

According to Appendix A, a total of 30 fatty acids were identified in the muscles of eels across different age groups, including 13 saturated fatty acids (SFAs), 7 monosaturated fatty acids (MUFAs), and 10 polyunsaturated fatty acids (PUFAs). Significant variations in fatty acid composition were observed among age groups. As the age increased, the proportion of SFAs decreased, while that of unsaturated fatty acids (UFA) increased. Notably, MUFAs were significantly lower than PUFAs in the 1-year-old group but surpassed PUFAs in older groups (3–22 years). The SFA profiles were consistent across age groups, dominated by C16:0 (20.99–23.95%) and C18:0 (4.55–5.68%). MUFAs primarily consisted of C18:1n9c (18.02–28.10%) and C16:1 (8.25–11.71%). In contrast, PUFA composition differed markedly between the 1-year-old group and older groups (3–22 years). The 1-year-old group exhibited an exceptionally high omega-3/omega-6 ratio of 1.58:1, with C18:3n3 being the predominant omega-3 PUFA. In older groups (3–22 years), this ratio declined to 0.34:1–0.78:1, and C22:6n3 (DHA) became the major omega-3 component.

### 3.2. Content of Amino Acids in Muscle

Appendix A presents the content of hydrolyzed amino acids (HAA) in muscle. A total of 16 HAAs were identified across different age groups, including 7 essential amino acids (EAAs, accounting for 50.714–70.496% of the total) and 9 non-essential amino acids (NEAAs, accounting for 29.504–49.286% of the total). Notably, the HAA content in the 3–11-year-old groups was significantly higher than in the 1-year and 22-year groups.

Appendix A presents the content of free amino acids (FAA) in muscle. A total of 20 FAAs were identified across different age groups, including 10 EAAs (accounting for 15.851–49.199% of the total) and 10 NEAAs (accounting for 50.801–84.149% of the total). In contrast to the content of HAAs, FAA levels in the 3–11-year-old groups were significantly lower than those in the 1-year and 22-year groups.

Figure 1 presents the content of flavor amino acids in muscle. Two umami amino acids (aspartic acid and glutamic acid) and five sweet amino acids (serine, proline, methionine, glycine, and alanine) were identified across different age groups. Their contents were significantly correlated with feeding age (*p* < 0.05). The content of total flavor amino acids, aspartic acid, glutamic acid, serine, and proline exhibited an initial increase followed by a decrease with advancing age, peaking in the age 3 group and reaching the lowest levels in the age 22 group. Conversely, the contents of methionine, glycine, and alanine showed an inverse pattern, with the highest values observed in the 1-year group and the lowest in the 11-year group.

### 3.3. Content of Gamma-Aminobutyric Acid, Triglycerides, and Cholesterol in Muscle

The contents of GABA, triglycerides, and cholesterol in muscle are illustrated in Figure 2. GABA was not detected in the 1-year and 22-year groups but was present in the 3–11-year groups (*p* < 0.05), with concentrations peaking in the 11-year group. Triglyceride content varied significantly across age groups (*p* < 0.05), particularly in the 22-year group, which exhibited markedly higher levels than other cohorts. The cholesterol content showed a trend of first increasing and then decreasing with age, with a significant increase in the 7-year group (*p* < 0.05). It is worth noting that the triglyceride content in the muscles of the 22-year group is the highest, while the cholesterol content is the lowest.

### 3.4. Principal Component Analysis of Muscle Nutritional Components

To explore age-related variations in nutritional composition, principal component analysis (PCA) was performed on fatty acids, amino acids, GABA, triglycerides, and cholesterol. As shown in Figure 3, PCA differentiated eel cohorts across five age groups (1, 3, 7, 11, and 22 years) along PC1 (82.53% variance) and PC2 (11.94% variance).

### 3.5. Muscle Texture Analysis

Figure 4 presents the results of the muscle texture analysis. With increasing age, hardness (*p* < 0.05), gumminess, and chewiness progressively decreased, whereas springiness (*p* < 0.05), cohesiveness, and resilience exhibited significant increases. Notably, the most pronounced age effects were observed in hardness reduction and resilience enhancement. Normalized textural parameters are visualized in the radar chart (Figure 5), revealing that the 1–3 year groups showed significantly higher hardness, chewiness, and gumminess compared to the 7–22 year groups. This trajectory suggests an age-dependent shift from rigid to resilient muscle architecture.

### 3.6. Slice Analysis

Figure 6 shows the histological structure of eel muscle across different age groups. In the 1- and 3-year groups, muscle fibers are tightly packed, structurally intact, and exhibit smaller diameters with uniform fiber spacing (Figure 6a,b). In contrast, the 7- and 11-year groups show enlarged fiber diameters, looser fiber arrangement, and reduced fiber spaces (Figure 6c,d). The 22-year group displays fiber diameters comparable to the 1- and 3-year cohorts but retains tight fiber alignment and uniform spacing (Figure 6e).

Figure 7 illustrates the analysis results of muscle fiber parameters in the slices. With advancing age, fiber diameter and average area initially increased significantly (*p* < 0.05), peaking in the 7-year group, followed by a marked decline (*p* < 0.05). Conversely, fiber density and number exhibited an inverse trend. Significant variations in fiber spacing were observed among age groups, with the 7- and 11-year groups showing the narrowest gaps compared to other cohorts (*p* < 0.05).

### 3.7. Correlation Between Muscle Texture and Muscle Fiber Parameters

To investigate the association between muscle texture and muscle fiber characteristics across different age groups, we conducted a Pearson correlation analysis between textural parameters (hardness, gumminess, chewiness) and morphometric fiber traits (density, number, diameter) in eels. The results are shown in Figure 8. Among muscle textural properties, hardness, gumminess, and chewiness exhibited a significant positive correlation with muscle fiber density and the number of muscle fibers (*p* < 0.01) while exhibiting a substantial negative correlation with muscle fiber diameter (*p* < 0.05). Negative correlation coefficients indicated that increased muscle fiber density and the number of muscle fibers were accompanied by a decrease in resilience (*p* < 0.05).

## 4. Discussion

### 4.1. The Effect of Different Rearing Years on Nutritional Composition in M. albus Muscle

This study systematically analyzed the dynamic changes in fatty acids, amino acids, GABA, triglycerides, and cholesterol content in eel muscles across different feeding ages. PCA revealed significant variations in their nutritional profiles related to growth age, similar to the phenomenon observed in poultry [34], edible insects [35,36], ginseng [37], and fruits [38].

Fatty acids, as an important indicator for evaluating meat quality, are also one of the important sources of nutrition and energy for the human body. Research has found that a diet rich in MUFAs can effectively reduce triglyceride concentrations and increase high-density lipoprotein levels. The composition and content of PUFAs reflect the nutritional quality of lipids, with higher PUFA content significantly enhancing meat aroma [39]. In this experiment, as eels aged, the proportion of SFAs in the muscles of eels decreased, while the proportion of UFAs increased. When over 1 year old, MUFAs became the dominant component. This shift suggests that 1-year-old eels rely on PUFAs to support their rapid growth demands, while adults utilize MUFAs as their primary energy reserve [40]. UFAs are known to improve cardiovascular health, metabolic function, inflammatory responses, and cognitive performance [41,42]. Nutritionists recommend a polyunsaturated-to-saturated fatty acid ratio (P:S) ≥ 0.4 [43], and an ω-6/ω-3 PUFA ratio <4 to mitigate risks of cancer and coronary heart disease [44]. Our results indicate that eels aged 1–22 years meet these criteria, offering superior nutritional benefits compared to beef, lamb, pork, and marine fish [45,46]. It is worth noting that the proportion of ω-3/ω-6 in 1-year-old eels is significantly higher than in other older groups, with α-linolenic acid (C18:3n3, ALA) as the predominant ω-3 PUFA. Diets rich in ω-3 fatty acids are associated with cardiovascular protection and atherosclerosis prevention [47]. Given typical diets’ ω-6 predominance, consuming 1-year-old eels (high ω-3/ω-6 ratio) could counterbalance this imbalance. In 7- to 22-year-old eels, docosahexaenoic acid (C22:6n3, DHA) becomes the dominant ω-3 component. DHA functions as a neurotrophic factor [48], modulates synaptic plasticity [49], and participates in anti-inflammatory pathways [50]. Therefore, older eels (≥7 years) are a viable source of DHA supplementation.

Amino acid analysis also highlights the impact of age on the nutritional value of eels. HAAs comprise a mixture of short peptides generated through protein hydrolysis and a minor proportion of FAAs [51]. These short peptides are rapidly absorbed via the intestinal peptide transport system [52], exhibiting higher absorption efficiency and lower osmotic pressure compared to FAAs [53], thereby making them ideal for infants or postoperative patients with compromised digestive function [54]. In addition, short peptides may retain bioactive properties, including antioxidant [55] and immune regulation [56]. FAAs, existing in monomeric form, require no enzymatic digestion for direct absorption, with bioavailability approaching 100%. This characteristic positions FAAs as precise supplements for targeted amino acid delivery [57]. HAAs complement each other in applications. Among them, aspartic acid can promote the production of neurotransmitters, glutamic acid can participate in human protein synthesis and enhance immunity, and lysine has the effect of promoting human protein utilization and protein synthesis [58,59]. In this study, HAA analysis revealed that EAAs accounted for 50.71–70.50% of total HAAs in the 3–11-year groups, significantly exceeding levels in the 1- and 22-year groups. This indicates that the protein quality of eel muscle is better at this stage, especially due to the enrichment of EAAs like lysine and leucine, which may enhance its potential as a high-quality protein source. Conversely, FAA distribution is inversely correlated with HAA trends. This dichotomy likely reflects metabolic adaptations: older eels (22-year group) may reduce energy expenditure by suppressing FAA levels while accumulating HAAs to sustain tissue functionality [60].

GABA, a naturally occurring non-protein amino acid, serves as the primary inhibitory neurotransmitter in the mammalian central nervous system. It regulates cardiovascular functions such as blood pressure and heart rate [61] and plays a role in reducing anxiety and pain [62,63]. In this study, eels aged 3–11 years showed elevated GABA levels, which further enhanced their health-promoting potential compared to other age cohorts.

Cholesterol, a key component of the cell membrane, maintains structural stability and serves as a precursor for synthesizing bile acids [64,65], vitamin D [66], and cortisol [67]. It also plays critical roles in nerve conduction and immune regulation [68,69]. Triglycerides, the primary form of energy storage, provide sustained fuel for muscular and organ functions [70]. While both cholesterol and triglycerides are essential lipids, their levels must be balanced within optimal physiological ranges to mitigate health risks [71]. In this study, the 22-year-old group exhibited peak triglyceride content alongside the lowest cholesterol levels. This lipid profile may confer dual advantages: enhanced palatability from lipid richness and reduced cardiovascular risks from low cholesterol. Coupled with elevated DHA and monounsaturated fatty acid levels, 22-year-old eels could be strategically positioned as premium functional ingredients that integrate health benefits with sensory appeal.

### 4.2. The Effect of Different Rearing Years on Flavor Quality in M. albus Muscle

In addition, this study systematically revealed the age-dependent characteristics of the flavor and taste of eel muscle. The dynamic changes in taste amino acid composition, texture characteristics, and muscle fiber structure jointly shape the differences in edible quality at different developmental stages.

From the perspective of flavor characteristics, aspartic acid and glutamic acid contribute umami taste to food, while serine, proline, threonine, glycine, and alanine impart sweetness [72]. Furthermore, flavor-enhancing amino acids synergize with taste-active nucleotides to amplify flavor complexity and intensity [73,74]. In this study, the concentrations of umami- and sweet-associated amino acids in eel muscle exhibited pronounced age-dependent divergence. Total flavor amino acids peaked in the 3-year-old group, primarily driven by the accumulation of aspartic acid and glutamic acid. In contrast, the 22-year-old group showed a marked reduction in umami amino acids (with a 40% reduction in glutamic acid content). Previous studies indicate that tissue amino acid profiles are modulated by developmental stages, likely reflecting adaptive responses to growth demands [75,76,77]. Based on the changes in amino acid content, the 3-year-old group may represent the “flavor golden window” for eels, where peak total flavor amino acids—via umami–sweetness synergy—align with consumer preferences for aquatic products. While the 1-year-old group exhibited the highest sweet amino acid levels, the delayed accumulation of umami components resulted in a unidimensional sweet profile, making it suitable for specific culinary applications (such as sweet-dominated sauce braised products).

The texture characteristics of fish meat, including hardness, cohesiveness, springiness, adhesiveness, gumminess, and chewiness, are critical determinants of consumer acceptance [78]. It is related to taste, tissue morphology, and other sensory qualities [79]. Hardness reflects resistance to mastication and is positively correlated with springiness; gumminess is defined as the product of hardness and cohesiveness; chewiness is calculated as the product of hardness, cohesiveness, and springiness; resilience quantifies the material’s capacity to recover shape post-deformation [80]. In this study, correlation patterns between textural parameters mirrored those reported in Atlantic salmon (*Salmo salar*) and brown trout (*Salmo trutta*) [81,82,83], but opposite to terrestrial animals [84,85]. Eels are benthic aquatic animals. We hypothesize that their muscles must adapt to continuous swimming and flexible body movements. This adaptation likely drives muscle fibers to arrange in a dense, numerous, and small-diameter pattern. This contrasts with terrestrial animals, where age-dependent textural changes primarily reflect load-bearing adaptations [86,87]. While feed composition [28,88] and aquaculture practices [16] are known modulators of piscine texture, this study provides novel evidence that the ontogenetic stage independently regulates textural properties. In this study, the muscle hardness and chewiness of the 1- and 3-year age groups were significantly higher than those of the 7–22 age groups, indicating that young eels have denser meat and a more elastic and tough taste, which may be related to the microstructural characteristics of tightly arranged dorsal muscle fibers and smaller diameters. In contrast, the muscle fibers of 7–11-year-old eels become loose, and the gaps shrink, leading to a decrease in hardness and chewiness; consequently, the meat tends to become soft. This change may be related to the shift in energy allocation toward reproductive development after sexual maturity [89]. This result reveals the unique regulatory mechanism of its muscle texture characteristics.

## 5. Conclusions

This study demonstrates that as a characteristic aquatic species, the nutritional and sensory properties of *Monopterus albus* muscle exhibit significant age-dependent variations. The 1-year-old eels exhibit a high ω-3/ω-6 ratio (1.58:1) and are enriched with sweet-tasting amino acids; the 3–11-year-old cohorts show prominent proportions of hydrolyzed EAAs; the 22-year-old eels demonstrate elevated DHA levels, reduced cholesterol content, and neuroactive metabolites. The age-dependent transition from stiff to elastic muscle texture correlates with muscle fiber morphometrics, contrasting sharply with the adaptations observed in terrestrial animals. The above findings provide a scientific basis for differentiated breeding and precision processing.

## Figures and Tables

**Figure 1 foods-14-01685-f001:**
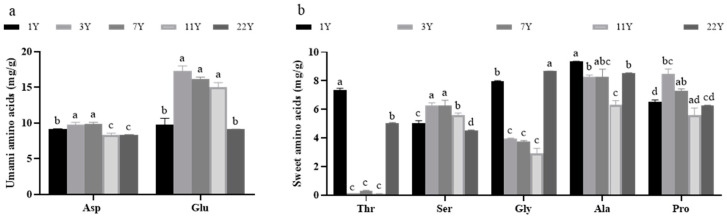
Effects of different rearing years on the content of flavor amino acids in *M. albus* muscle. (**a**) the content of umami amino acids; (**b**) the content of sweet amino acids. Different lowercase letters indicate significant differences among the groups (*p* < 0.05).

**Figure 2 foods-14-01685-f002:**
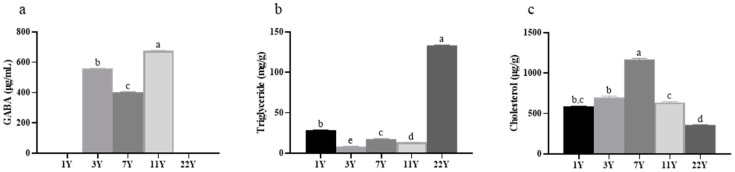
Effects of different rearing years on the content of (**a**) γ-aminobutyric acid, (**b**) triglyceride, and (**c**) cholesterol in *M. albus* muscle. Different lowercase letters indicate significant differences among the groups (*p* < 0.05).

**Figure 3 foods-14-01685-f003:**
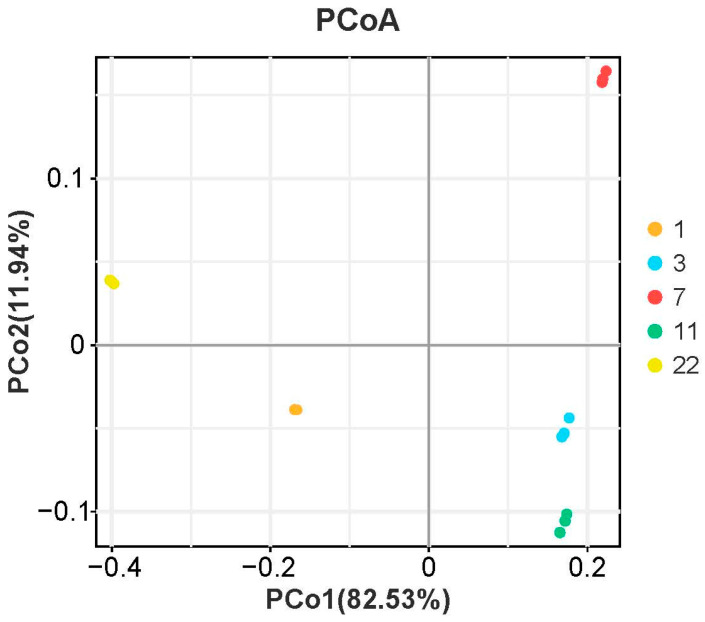
Principal coordinates analysis (PCoA) analysis of nutritional components in *M. albus* muscle of different feeding ages.

**Figure 4 foods-14-01685-f004:**
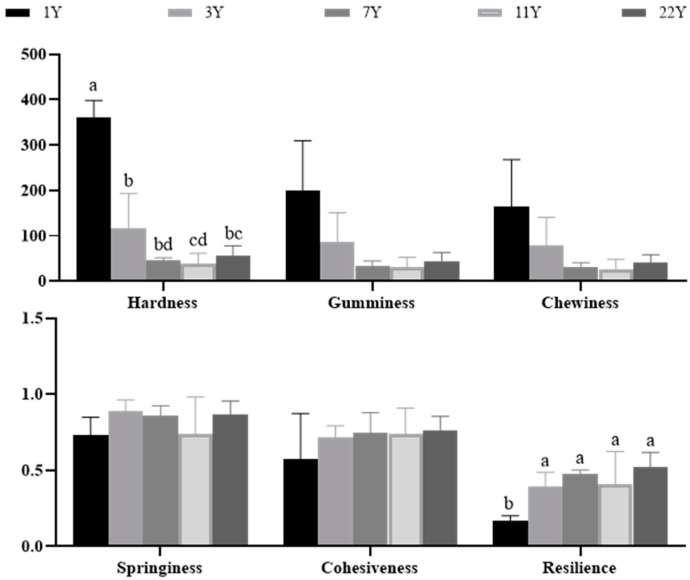
Effects of different rearing years on texture profile in *M. albus* muscle. Different lowercase letters indicate significant differences among the groups (*p* < 0.05).

**Figure 5 foods-14-01685-f005:**
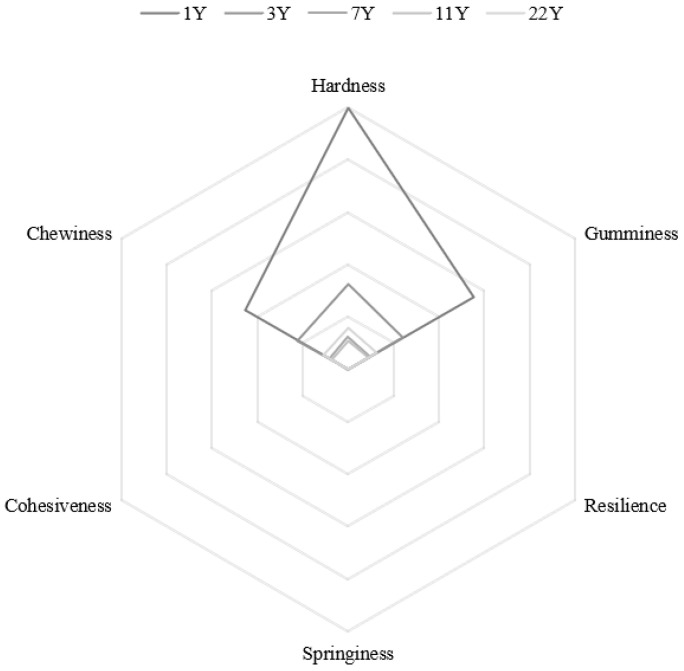
Radar chart of different feeding ages *M. albus* muscle texture parameters.

**Figure 6 foods-14-01685-f006:**
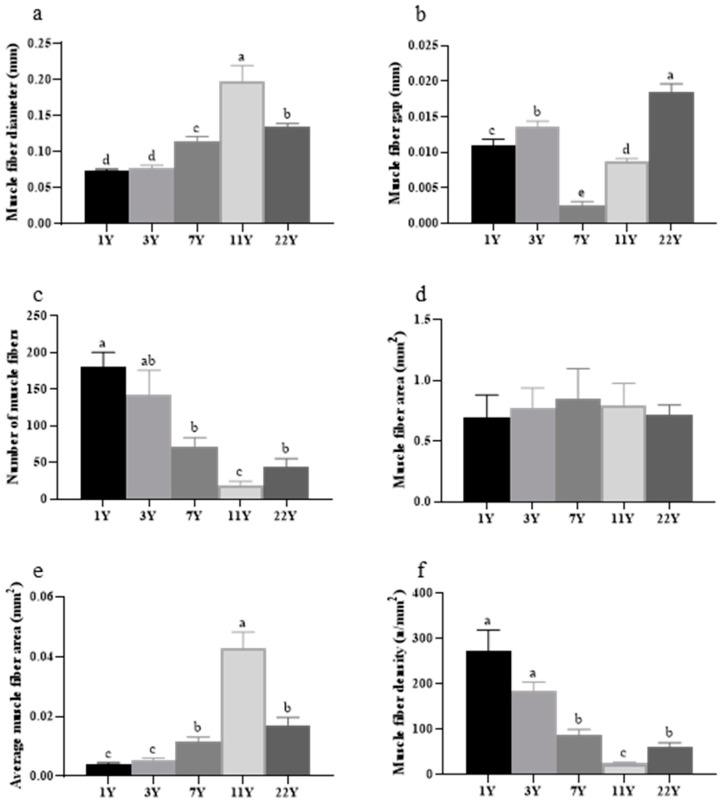
Measurement and analysis of back muscle cells in *M. albus* muscle of different feeding ages. (**a**) the diameter of muscle fibers; (**b**) the spacing between muscle fibers; (**c**) the number of muscle fibers; (**d**) the area of muscle fibers; (**e**) the average area of muscle fibers; (**f**) the density of muscle fibers. Different lowercase letters indicate significant differences among the groups (*p* < 0.05).

**Figure 7 foods-14-01685-f007:**
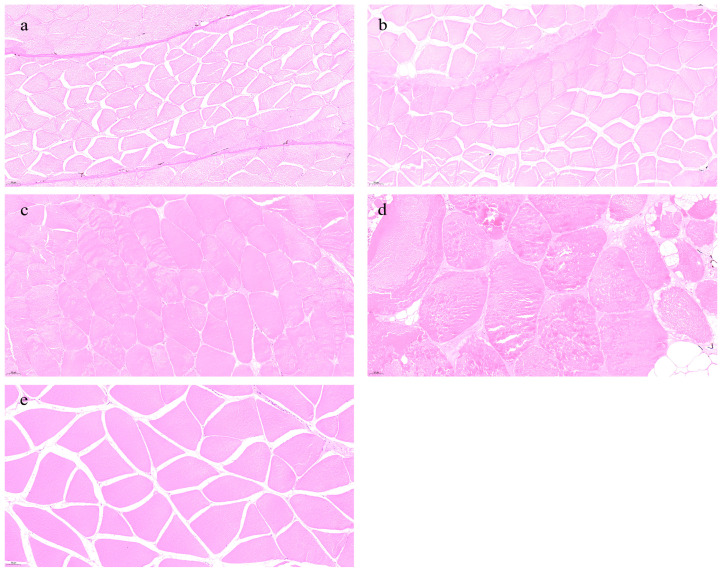
Effects of different rearing years on morphological changes in *M. albus* muscle. (**a**) age 1, (**b**) age 3, (**c**) age 7, (**d**) age 11, and (**e**) age 22.

**Figure 8 foods-14-01685-f008:**
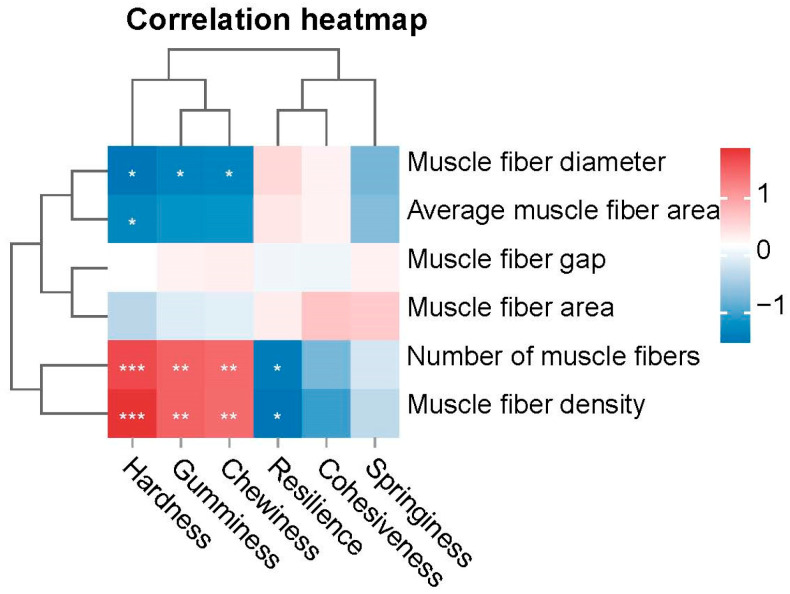
Correlation Between Muscle Texture and Muscle Fiber Parameters. The asterisk indicates the significant difference in *p*-value between groups, * indicates *p* < 0.05, ** indicates *p* < 0.01, *** indicates *p* < 0.001.

## Data Availability

The original contributions presented in the study are included in the article/Appendix A. Further inquiries can be directed to the corresponding authors.

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
