# Peer review of "Impact of Rearing Duration on Nutritional Composition, Flavor Characteristics, and Physical Properties of Asian Swamp Eel (Monopterus albus)"

_foods, 2025, doi:10.3390/foods14101685_

Round 1
Reviewer 1 Report
Comments and Suggestions for Authors
Dear author, the revised manuscript is interesting. In this initial review, the following recommendations are made:
Line 27: Insert the meaning of DHA and GABA
Line 29: Rewrite… ingredients. The
Line 32: It is commonly recommended that some of the keywords be different from those found in the title to increase the document's visibility during a database search.
Line 32: Use semicolons instead of commas to separate keywords
Line 35: Could you include information on the production and consumption of this species in a specific region?
Line 36: Could indicate the percentage range of the majority components.
Line 48: Insert the meaning of PUFAs
Line 48: Use italic text format for scientific names
Line 51: Did you mean Culter alburnus?
Line 53: What is the meaning of ΣEAA?
Line 91: Add the approval number or reference
Line 92: A section on chemicals and reagents, including all those used in the study, their origin, and purity, must be added.
Line 111: All procedures in this section correspond to reference 29?
Line 115: It is recommended that authors' names or surnames be avoided in the text; only the reference number that corresponds to the information should be used, and this should be placed in brackets.
Line 130: Idem
Line 142: The abbreviation was previously used for some terms in this line; when a term or word is abbreviated for the first time, this abbreviation must be used subsequently throughout the document.
Line 159: You can indicate in the title that the results are expressed in % and eliminate the symbol in the values ​​in the table.
Line 172,184: It is required to add the meaning of the abbreviations of the information contained in the table
Line 222: In the analysis of the principal components, the separation or grouping of the individual nutritional components is not observed; only the separation of the samples in time is observed. It is necessary to include the analyzed nutrients to see how they are grouped according to the age groups.
Line 279: rewrite… 4.1. The Effect of Different Rearing Years on Nutritional Composition in M. albus Muscle
Line 353: rewrite… 4.2. The Effect of Different Rearing Years on Flavor Quality in M. albus Muscle
Line 377: remove text space… appeal,
Line 425: Carefully review the text format in this section.
Author Response
Responses to reviewer #1:
Thank you very much for your support and affirmation of this paper. According to your constructive and detailed advices, we have carefully revised our manuscript, and the revision was shown below. We believe your advices will improve our paper.
Comments 1: Line 27: Insert the meaning of DHA and GABA
Response: Thanks for your detailed suggestion. The meanings of DHA and GABA have been inserted on line 25.
Comments 2: Line 29: Rewrite… ingredients. The
Response: This area has been corrected.
Comments 3: Line 32: It is commonly recommended that some of the keywords be different from those found in the title to increase the document's visibility during a database search.
Response: The keywords in this manuscript have been modified to “Asian eel; nutritional composition; texture analysis; flavor profiles”.
Comments 4: Line 32: Use semicolons instead of commas to separate keywords
Response: This area has been corrected.
Comments 5: Line 35: Could you include information on the production and consumption of this species in a specific region?
Response: Thank you for your proposal. The latest production information of eels has been added to line 35 of the manuscript.
Comments 6: Line 36: Could indicate the percentage range of the majority components.
Response: This section has been revised to address the reviewers' comments. We hope it aligns with your expectations.
Comments 7: Line 48: Insert the meaning of PUFAs
Response: The meanings of PUFAs has been inserted on line 48.
Comments 8: Line 48: Use italic text format for scientific names
Response: The font has been modified.
Comments 9: Line 51: Did you mean Culter alburnus?
Response: Yes, thank you for your friendly reminder. The scientific names have been changed to italics.
Comments 10: Line 53: What is the meaning of ΣEAA?
Response: The meanings of ΣEAA has been inserted on line 53.
Comments 11: Line 91: Add the approval number or reference
Response: Approval number SAASXM062438 has been added on line 86.
Comments 12: Line 92: A section on chemicals and reagents, including all those used in the study, their origin, and purity, must be added.
Response: The information on chemical drugs and reagents has been added on line 87-97.
Comments 13: Line 111: All procedures in this section correspond to reference 29?
Response: The analysis of muscle amino acid profile refers to this literature. The content of this paragraph has been modified to clarify the expression.
Comments 14: Line 115: It is recommended that authors' names or surnames be avoided in the text; only the reference number that corresponds to the information should be used, and this should be placed in brackets.
Response: Corrected.
Comments 15: Line 130: Idem
Response: Corrected.
Comments 16: Line 142: The abbreviation was previously used for some terms in this line; when a term or word is abbreviated for the first time, this abbreviation must be used subsequently throughout the document.
Response: The abbreviations of terms in the manuscript have been thoroughly checked to ensure that they are explained when they first appear.
Comments 17: Line 159: You can indicate in the title that the results are expressed in % and eliminate the symbol in the values ​​in the table.
Response: Corrected.
Comments 18: Line 172,184: It is required to add the meaning of the abbreviations of the information contained in the table
Response: Corrected.
Comments 19: Line 222: In the analysis of the principal components, the separation or grouping of the individual nutritional components is not observed; only the separation of the samples in time is observed. It is necessary to include the analyzed nutrients to see how they are grouped according to the age groups.
Response: Thanks for your comments. This study mainly focuses on the effects of different feeding years on the nutritional composition and flavor quality of fish meat. In order to explore the age-related changes in the detected nutritional components, fatty acids, amino acids, GABA, Triglycerides and cholesterol were subjected to PCA, with the hope of obtaining comprehensive findings in principal component analysis. Age is the independent variable in this study, therefore the separation of samples over time is presented in Figure 3. The analysis of individual nutritional components can be intuitively observed from the existing charts in the article, so we think that further principal component analysis may be not necessary.
Comments 20: Line 279: rewrite… 4.1. The Effect of Different Rearing Years on Nutritional Composition in M. albus Muscle
Response: Corrected.
Comments 21: Line 353: rewrite… 4.2. The Effect of Different Rearing Years on Flavor Quality in M. albus Muscle
Response: Corrected.
Comments 22: Line 377: remove text space… appeal,
Response: Corrected.
Comments 23: Line 425: Carefully review the text format in this section.
Response: Corrected.
Reviewer 2 Report
Comments and Suggestions for Authors
Impact of Rearing Duration on Nutritional Composition, Flavor Characteristics and Physical Properties of Asian Swamp Eel (Monopterus albus)
OBJECTIVE: This study aims to systematically analyze the dynamic changes in nutritional components, flavor compounds, and muscle texture of eel muscles across five feeding durations (1, 3, 7, 11,22 years)
- L16-20: The content of this phrase is important in the abstract, but it’s a bit too long and a Little confusing. Please rewrite.
- L27: What is “balanced nutrition”? Maybe the authors explain this term in further sections, but in the abstract it should be explained.
- L43: Typo in weight
- L48: Scientific names in italics
- L51: Scientific names in italics
- L53: I suggest that the meaning of essential amino acids (EAA) should be stated, as it is the first time it appears in the text
- L61-62: Texture is mentioned twice, and it causes the phrase to lose sense
- L90: More information on the ethics committee must be provided (for instance, registration number)
- L95: In my opinion, the symbols for meters and square meters is better (m, m2)
- L108: Grammatical error “Fix the sample slices with 5% paraformaldehyde”
- L125: Elasticity is repeated
- L242: The word “clearly” is unnecessary
- L283: Is it correct to say “feeding age” when including ginseng and fruits?
- L338: “Blood pressure” as it is, it’s not a disease. High or low blood pressure can be considered a cardiovascular-related disease.
- L341-345: This information could be removed since, by now, it’s common knowledge, but I leave it up to the authors.
- L377: Typo
- L400: Conclusions are not a summary of results. I strongly recommend that the authors rewrite this section, making emphasis on their findings and their importance in the development of nutritious alternatives from marine sources, such as eels.
It presents only a few minor issues that require attention, such as typos, composition or formatting corrections.
Author Response
Responses to reviewer #2:
Thank you for your high evaluation and constructive and valuable questions. We believe that these questions will improve our manuscript better. The answers to your questions were listed below.
Comments 1: L16-20: The content of this phrase is important in the abstract, but it’s a bit too long and a Little confusing. Please rewrite.
Response: Thanks for your detailed suggestion. The abstract has been revised to clarify the expression. We hope it aligns with your expectations.
Comments 2: L27: What is “balanced nutrition”? Maybe the authors explain this term in further sections, but in the abstract it should be explained.
Response: The abstract has been revised to clarify the expression. We hope it aligns with your expectations.
Comments 3: L43: Typo in weight
Response: Corrected.
Comments 4: L48: Scientific names in italics
Response: Thank you for your friendly reminder. The scientific names have been changed to italics.
Comments 5: L51: Scientific names in italics
Response: Corrected.
Comments 6: L53: I suggest that the meaning of essential amino acids (EAA) should be stated, as it is the first time it appears in the text
Response: The meanings of ΣEAA has been inserted on line 53.
Comments 7: L61-62: Texture is mentioned twice, and it causes the phrase to lose sense
Response: Sorry for the typo, it has been corrected.
Comments 8: L90: More information on the ethics committee must be provided (for instance, registration number)
Response: Approval number SAASXM062438 has been added on line 86.
Comments 9: L95: In my opinion, the symbols for meters and square meters is better (m, m2)
Response: Corrected.
Comments 10: L108: Grammatical error “Fix the sample slices with 5% paraformaldehyde”
Response: This area has been corrected to: Muscle slices were fixed in 5% paraformaldehyde (PFA) for 24 h at 4°C.
Comments 11: L125: Elasticity is repeated
Response: Sorry for the typo, it has been corrected.
Comments 12: L242: The word “clearly” is unnecessary
Response: Deleted.
Comments 13: L283: Is it correct to say “feeding age” when including ginseng and fruits?
Response: Thanks for your comments. The wording has been changed to “PCA revealed significant growth age-related variations in their nutritional profiles, which is similar to the phenomenon observed in poultry [31], edible insects [32,33], ginseng [34], and fruits”.
Comments 14: L338: “Blood pressure” as it is, it’s not a disease. High or low blood pressure can be considered a cardiovascular-related disease.
Response: Thanks for your comments. The wording has been changed to “It regulates cardiovascular functions such as blood pressure and heart rate [58], and plays a role in reducing anxiety and pain”.
Comments 15: L341-345: This information could be removed since, by now, it’s common knowledge, but I leave it up to the authors.
Response: Thanks for your comments. This section has been deleted from the manuscript.
Comments 16: L377: Typo
Response: Corrected.
Comments 17: L400: Conclusions are not a summary of results. I strongly recommend that the authors rewrite this section, making emphasis on their findings and their importance in the development of nutritious alternatives from marine sources, such as eels.
Response: Thank you for your proposal. The conclusion chapter of this study has been rewritten on line 368-376. We hope that the new conclusions can emphasize the importance of the nutritional and sensory properties of Monopterus albus.
Comments 18: Comments on the Quality of English Language
It presents only a few minor issues that require attention, such as typos, composition or formatting corrections.
Response: Thank you for your reminder. The spelling, grammar, and formatting in the new manuscript have been thoroughly checked.
Reviewer 3 Report
Comments and Suggestions for Authors
The article "Impact of Rearing Duration on Nutritional Composition, Flavor Characteristics and Physical Properties of Asian Swamp Eel (Monopterus albus)" has a good experimental design and presents data relevant to the field of aquaculture and food science. However, for a rigorous evaluation, the following are the main strengths, serious problems.
Strengths:
- Clear scientific objective: The study effectively investigates the impact of rearing duration on muscle quality, nutrition, and flavor in Monopterus albus.
- Solid methodology: Utilizes GC, HPLC, histological and texture analysis with proper control of variables.
- Robust data presentation: Well-structured tables and figures, with appropriate statistical methods.
- Relevant discussion: Incorporates current literature and provides plausible physiological explanations for observed results.
1. Language and Grammar Issues (Major)
The manuscript contains frequent grammatical errors, awkward phrasing, and incorrect use of tense and pluralization.
A full professional English language editing is mandatory.
2. Structural Issues
Figures and tables, while rich in data, are visually dense. Long tables (e.g., fatty acid and amino acid profiles) should be considered for supplementary material.
3. Justification of Experimental Design
The use of 22-year-old eels is highly unusual in aquaculture. A stronger justification for including such an extended rearing period is necessary.
- “During the experiment, the dissolved oxygen level exceeded 6 mg/L, the pH range was 7.5 to 8.0, the ammonia nitrogen level was below 0.2 mg/L, and the temperature range was 30±2°C” — these data are presented in the Experimental Management section; however, they are in fact results, and I noticed there is no discussion about these parameters in relation to the study’s objective, which raised some concerns. Could these parameters, for instance, have influenced the nutritional components, flavor characteristics, and the physical and textural properties of the muscle?
- Results
Place metric bar on histological images.
- Discussion
Mixed usage of “young”, “adult”, “elderly” for eels—clarify age groups consistently.
- Weak Conclusion
The conclusion lacks synthesis and reiterates discussion points rather than summarizing the study’s main contributions and practical implications.
Study suggestion for contextualization in the introduction: ENRICHED CEREAL BAR WITH RAY MEAT Hypanus guttatus (DASYATIDAE – MYLIOBATIFORMES) CAPTURED ON MARANHÃO STATE COASTAL - https://portaldeperiodicos.animaeducacao.com.br/index.php/gestao_ambiental/article/view/9185
Reinforces the importance of evaluating nutritional and sensory properties of unconventional aquatic species such as Monopterus albus.
Comments on the Quality of English Language- "Hydrolyzed essential amino acids were enriched in 3–11-year groups" → "The levels of hydrolyzed essential amino acids were higher in the 3–11-year-old groups."
- Terms like “gelatinization” are incorrectly used in texture profile context.
Author Response
Responses to reviewer #3:
Thank you very much for your support and affirmation of this paper. According to your constructive and detailed advices, we have carefully revised our manuscript, and the revision was shown below. We believe your advices will improve our paper.
Comments 1: The manuscript contains frequent grammatical errors, awkward phrasing, and incorrect use of tense and pluralization.
Response: Thank you for your reminder. The spelling, grammar, and formatting in the new manuscript have been thoroughly checked. The language of the manuscript has been polished by native English speakers, and we hope that the revised manuscript can meet the quality requirements of the journal.
Comments 2: Figures and tables, while rich in data, are visually dense. Long tables (e.g., fatty acid and amino acid profiles) should be considered for supplementary material.
Response: Thank you for your friendly reminder. The fatty acid and amino acid profiles have been changed to supplementary materials
Comments 3: The use of 22-year-old eels is highly unusual in aquaculture. A stronger justification for including such an extended rearing period is necessary.
Response: Eels can have a lifespan of 6-7 years in their natural state, and under good feeding conditions, their lifespan is even longer. The establishment of the 22 year old eel group is not based on conventional breeding practices, but rather serves as an experimental model to explore the biological limits of the species and the potential for industrial innovation. The setting of the 22 year old group can fill the research gap in the nutritional composition and flavor quality of eels at extreme ages.
Comments 4: “During the experiment, the dissolved oxygen level exceeded 6 mg/L, the pH range was 7.5 to 8.0, the ammonia nitrogen level was below 0.2 mg/L, and the temperature range was 30±2°C” — these data are presented in the Experimental Management section; however, they are in fact results, and I noticed there is no discussion about these parameters in relation to the study’s objective, which raised some concerns. Could these parameters, for instance, have influenced the nutritional components, flavor characteristics, and the physical and textural properties of the muscle?
Response: Thanks for your comments. This study mainly focuses on the effects of different feeding years on the nutritional composition and flavor quality of fish meat. We used age as the independent variable in this study by controlling for variables. Therefore, the same feeding and management methods will not significantly affect the nutritional composition, flavor characteristics, as well as the physical and texture of muscles. So, we think there is no need to discuss this matter further.
Comments 5: Place metric bar on histological images.
Response: The bottom left corner of each slice is marked with a ruler.
Comments 6: Mixed usage of “young”, “adult”, “elderly” for eels—clarify age groups consistently.
Response: To avoid confusion about the age of eels caused by the use of "young", "adult", and "elderly", specific age groups are used throughout the text.
Comments 7: The conclusion lacks synthesis and reiterates discussion points rather than summarizing the study’s main contributions and practical implications. Reinforces the importance of evaluating nutritional and sensory properties of unconventional aquatic species such as Monopterus albus.
Response: Thank you for your proposal. The conclusion chapter of this study has been rewritten on line 368-376. We hope that the new conclusions can emphasize the importance of the nutritional and sensory properties of Monopterus albus.
Comments 8: "Hydrolyzed essential amino acids were enriched in 3–11-year groups" → "The levels of hydrolyzed essential amino acids were higher in the 3–11-year-old groups."
Response: Corrected.
Comments 9: Terms like “gelatinization” are incorrectly used in texture profile context.
Response: Corrected.
Comments 10: Study suggestion for contextualization in the introduction: ENRICHED CEREAL BAR WITH RAY MEAT Hypanus guttatus (DASYATIDAE – MYLIOBATIFORMES) CAPTURED ON MARANHÃO STATE COASTAL - https://portaldeperiodicos.animaeducacao.com.br/index.php/gestao_ambiental/article/view/9185
Response: In the introduction of the manuscript, we did not find a suitable context to cite the conclusion of the article. We still greatly appreciate your recommendation, and if there is a suitable opportunity, we will cite the article.
Round 2
Reviewer 1 Report
Comments and Suggestions for Authors
Dear authors, the recommendations from the first revision of the document have been addressed.
Reviewer 3 Report
Comments and Suggestions for Authors
It's ok.